# Colorless and Transparent Polyimide Microporous Film with Excellent Physicochemical Property

**DOI:** 10.3390/polym13081298

**Published:** 2021-04-15

**Authors:** Jong Won Kim, Seon Ju Lee, Moon Young Choi, Jin-Hae Chang

**Affiliations:** 1Department of Polymer Science and Engineering, Kumoh National Institute of Technology, Gumi 39177, Korea; kiw010@naver.com; 2Graduate School of Carbon Convergence Engineering, Jeonju University, Jeonju 55069, Korea; qaz5515@naver.com (S.J.L.); ansdud2303@naver.com (M.Y.C.); 3Institute of Carbon Technology, Jeonju University, Jeonju 55069, Korea

**Keywords:** colorless and transparent polyimide, poly(vinyl alcohol), blend, microporous film, physicochemical property

## Abstract

4,4′-(4,4′-isopropylidenediphenoxy)bis(phthalic anhydride) (BPADA) as a dianhydride and bis(3-aminophenyl) sulfone (APS) and bis(3-amino-4-hydroxyphenyl) sulfone (APS-OH) as diamines were used to synthesize two types of poly(amic acid) (PAA). Varying amounts (0–5.0 wt%) of water-soluble poly(vinyl alcohol) (PVA) were mixed with PAA, and the resulting blend was heat-treated at different stages to obtain the colorless and transparent polyimide (CPI) blend films. The synthesized blended film completely removed water-soluble PVA in water. The possibility as a porous membrane according to the pore size varied according to the amount of PVA was investigated. The dispersibility and compatibility of CPI containing APS-OH monomer were higher than those of the APS monomer. This could be attributed to the hydrogen-bonding interactions between the CPI main chains and PVA. Scanning electron microscopy was conducted to characterize the material. The results revealed that the pore size of the CPI blend film increased as the PVA concentration increased. It was confirmed that uniform pores of μm-size were observed in CPI. The thermal stabilities, morphologies, optical properties, and solubilities of two CPIs obtained using APS and APS-OH monomers were investigated and their properties were compared with each other.

## 1. Introduction

Many impurities present in industrial wastewater and domestic drinking water are difficult to remove because of their diverse and complex structures. Methods of treating contaminated solvents and wastewater include the use of simple filters, agglomeration, powdering, biological drainage, and ozonation. In addition, activated carbon adsorption, photocatalytic, and electrochemical processes have been used. However, it is difficult to completely remove impurities using these methods. In addition, most of these methods are more contaminated by impurities generated during refining [1,2,3]. 

Various types of membrane filters, including microfiltration (MF), nanofiltration (NF), hollow fiber membrane filtration (UF), and reverse osmosis (RO), have been applied to wastewater refinement, industrial fluids, and dye treatment. This filtration process is designed to filter a wide variety of materials [4,5]. Depending on the pore size, MF, UF, NF, and RO can separate small particles, macromolecules, nanomaterials, and ionic compounds, respectively. Therefore, the main water purification technology, MF was used to eliminate bacteria and various suspended substances, while UF was used to exclude colloids and various viruses. NF was used to remove heavy metals and organic substances from solvents, and RO was applied mainly in desalination, water reprocessing facilities, and household water purifiers. This method has been widely used in various fields and reported in many studies [6,7].

MF was first commercialized in 1929 by Sartorius Werke GmbH in Göttingen, Germany, to reliably supply drinking water by analyzing bacteria in water samples from several German cities bombed during World War II [8]. However, the first patent on a microporous membrane was registered by Zsigmondy in 1922 [9]. MF is a membrane process driven by pressure on a microporous membrane, removing suspended colloids and particles in the size range 0.1–20 mm. MF is generally operated at a relatively low pressure (50 psi/3.4 bar/0.35 MPa) to separate impurities, but with an uncontaminated membrane, it operates at very high permeation rates (10^−4^–10^−2^ m/s). These operating conditions make MF different from RO and UF [8]. Meanwhile, the characterizations and applications of various MF membranes, manufactured for water purification, have been demonstrated. Zhong et al. [10] developed a ceramic MF membrane, containing zirconium dioxide (ZrO_2_), with excellent water treatment properties. Previous MF membrane studies have shown that inorganic membranes are substances effective in removing contamination through pretreatment steps, such as agglomeration [11,12].

Polyimide (PI) can be easily processed into membranes with high chemical and physical stability. PI membranes can be used under various conditions due to the unique properties of PI and have excellent separation properties for gases and liquids. In addition, PI showed various potential as a microporous film with excellent chemical resistance, thermo-mechanical properties, and processability [13,14]. 

PI has a characteristic brown color that absorbs light in the visible region through the transition of numerous π electrons present in the main chain to the adjacent chain, owing to the characteristics of the aromatic structure. To remove such a brown color, a colorless and transparent PI (CPI) can be formed by introducing a monomer with a curved structure into the main chain, or by using a fluorine- (-F, -CF_3_) or sulfone-substituted (-SO_2_-) monomer with high electronegativity. CPI has superior thermal stability, solubility, and optical transparency compared to conventional, colored PIs. In addition, CPI films have many potential applications in electro-optical devices, especially in semiconductor applications, including flexible display devices [15,16].

Poly(vinyl alcohol) (PVA) is the most widely known water-soluble polymer, but it is not produced by the direct reaction of the corresponding monomer, which is spontaneously converted thermodynamically to the enol form of acetaldehyde. It is commercially produced through the hydrolysis of poly(vinyl acetate). PVA is widely used for flexible, water-soluble packaging films, coatings, and textile sizing [17]. PVA contributed to improving the thermo-mechanical properties of a blend film consisting of various polymers, while maintaining its optical transparency. Additionally, various polymeric materials containing PVA can be applied not only as blends but also as substitutes for other polymer materials in special fields with high functionality [18,19,20]. 

Compounds containing -OH allow strong intermolecular hydrogen bonding, an electrostatic attraction that occurs between atoms with high electronegativity, such as O, N, F, and H atoms in neighboring molecules attached to a similar electronegative atom (O, N, or F), and is a type of intermolecular attraction. In addition, hydrogen bonds are considered relatively strong among the typical intermolecular forces [21].

In our study, the main chain structure of CPI was designed to include an -OH group, and its properties were compared to those of the general CPI structure without the -OH group. If hydrophilic CPI with -OH groups is blended with PVA to form hydrogen bonds, we expect the blends to exhibit excellent dispersibility and compatibility through hydrogen bonding.

The purpose of this study was to synthesize CPI blend films with various pore sizes. To this end, a poly(amic acid) (PAA)/PVA blend film was synthesized by blending PVA in the range 0–5 wt% with the synthesized PAA, and, after synthesizing a CPI/PVA blend film through various heat treatment processes, a CPI microporous film was prepared by removing the water-soluble PVA with an aqueous solution. In addition, this study investigated the potential of CPI for membrane fabrication by improving the thermal properties, optical transparencies, and chemical resistances. We also focused on controlling the μm-size of the CPI membrane using a simple method of dissolving water-soluble PVA in water.

## 2. Materials and Methods

### 2.1. Materials

Bis(3-aminophenyl) sulfone (APS), bis(3-amino-4-hydroxyphenyl) sulfone (APS-OH), and 4,4′-(4,4′-isopropylidenediphenoxy)bis(phthalic anhydride) (BPADA) were purchased from Santa Cruz Biotechnology (Shanghai, China) and Tokyo Chemical Industry (Tokyo, Japan). *N,N*-dimethylacetamide (DMAc) was obtained from Junsei Chemical Co., Ltd. (Tokyo, Japan), and moisture was completely removed with molecular sieves (4 Å). PVA obtained through an 80% degree of saponification (DP = 2.04 × 10^2^) was obtained from Aldrich Chemical Co (Yongin, Korea). All other solvents were used without further purification.

### 2.2. Methods

The methods for synthesizing CPI using two monomers, APS and APS-OH, were identical. For example, the synthesis using APS proceeded as follows: PAA was obtained from BPADA and APS, using DMAc. BPADA (6.92 g; 1.3 × 10^−2^ mol) and 40 mL of DMAc were mixed in a beaker at 0 °C for 30 min under N_2_ atmosphere. In a separate beaker, DMAc (20 mL) was mixed with APS (3.30 g; 1.3 × 10^−2^ mol), and this solution was then mixed with the prepared BPADA solution.

For stabilization under N_2_ condition, the solution was slowly stirred at 25 °C for 1 h, and then at 0 °C for 1 h, and then at 25 °C for 12 h to prepare a PAA solution. To a three-necked flask with a reflux condenser, PVA (0.1 g; 2.3 × 10^−2^ mol), 1 wt% PAA solid content, was supplemented to the PAA, followed by mixing under N_2_ atmosphere at 50 °C for 3 h. The PAA/PVA blend was coated on a glass panel to a certain thickness using a coating bar, and the solvent was slowly eliminated while stabilizing the PAA for 2 h at 50 °C in a vacuum oven. The solvent was then completely removed under vacuum at 80 °C for 1 h to prepare blended PAA/PVA films. The thermal imidization reaction to obtain the CPI/PVA blend film was maintained at 110, 140, and 170 °C under vacuum for 30 min, and then heat-treated at 195 and 220 °C for 50 min. After all, heat imidization was finished by holding the mixture at 235 °C for 2 h. The detailed reaction conditions are listed in Table 1. After completion of the imide reaction, the CPI/PVA film was slowly peeled off the glass panel in water at 90 °C. The overall synthetic route is illustrated in Scheme 1.

### 2.3. Synthesis of Microporous CPI Film 

Because PVA is a hydrophilic polymer, it is completely soluble in hot water. Therefore, when the CPI/PVA blend film was immersed in water at 90 °C for 5 h to completely dissolve the PVA component in the blend film, it was possible to prepare porous CPI membrane films with the desired pore sizes, according to various amounts of PVA (wt%).

### 2.4. Characterization

The structure of the synthesized material was confirmed using Fourier-transform infrared (FT-IR) spectroscopy (Bruker VERTEX 80v, Berlin, Germany) and Carbon-13 nuclear magnetic resonance (^13^C-NMR) spectroscopy (Bruker 400 DSX NMR, Berlin, Germany). The ^13^C cross-polarization (CP)/magic-angle spinning (MAS) NMR experiment was conducted at a Larmor frequency of 100.61 MHz. To minimize the spinning sideband, it was measured at an MAS rate of 12 kHz. Tetramethylsilane (TMS) was used as a standard to record the NMR spectra. 

The thermal properties were investigated by differential scanning calorimetry (DSC, NETZSCH 200F3, Berlin, Germany) and thermogravimetric analysis (TGA, TA Instruments TA Q500, New Castle, DE, USA) under N_2_ atmosphere. The rate of temperature increase was maintained at 20 °C/min.

The degree of crystallinity (DC) of the film was measured with an X-ray diffractometer (XRD, Rigaku XRD, Tokyo, Japan) with Cu Kα radiation. Measurements were carried out at a scan rate step size of 2°/min in the 2θ range 2°–40°. The pore size of the films was investigated using a field emission scanning electron microscope (SEM, JEOL JSM-6500F, Tokyo, Japan). To investigate the optical transparencies, a spectrophotometer (Konica Minolta CM-3600D, Tokyo, Japan) and a UV-visible (UV-vis) spectrophotometer (Shimadzu UV-3600, Tokyo, Japan) were used.

## 3. Results and Discussion

### 3.1. FT-IR

The FT-IR spectra of the CPI films synthesized using APS or APS-OH monomers are shown in Figure 1. For the APS CPI film (Figure 1a), a C=O aromatic stretching peak was observed at 1778 cm^−1^ and 1718 cm^−1^, and a C–N–C peak, representing an imide cyclization, was shown at 1360 cm^−1^. However, for the APS-OH CPI film (Figure 1b), a wide O-H stretching peak was shown at ~3300 cm^−1^. Since the -OH groups present in CPI are capable of hydrogen bonding, the intensity of the -OH elongation peak appearing on the CPI film is lower than that of the commonly observed hydroxyl peak. Molecules with -OH functional groups are capable of intramolecular hydrogen bonding between CPI main chains and represent a broad elongation absorption peak through a wide wavenumber range (3000–3300 cm^−1^). Therefore, the APS-OH PI spectrum shows a broad peak due to hydrogen bonding between the nitrogen and the -OH group of phenol in the adjacent imide ring as shown in Figure 1 [22,23,24]. Similar to the APS CPI spectrum, the APS-OH CPI spectrum showed a C=O aromatic stretching peak a C-N-C peak indicating imidization at 1777 cm^−1^, 1716 cm^−1^, and 1380 cm^−1^, respectively. It was found that the two imide reactions were completed [25,26,27].

### 3.2. Solid State C-13 NMR

The structural analyses of the synthesized APS CPI and APS-OH CPI was performed using solid-state ^13^C CP/MAS NMR, and the chemical shifts of ^13^C at room temperature are shown in Figure 2a,b. The ^13^C MAS NMR spectrum of TMS was obtained at 38.3 ppm at 25 °C. This peak at 38.3 ppm was taken as the standard and calibrated as a peak at 0 ppm. 

In the APS CPI spectrum (Figure 2a), the chemical shifts for the methyl (-CH_3_) and isopropyl carbons (-C-(CH_3_)_2_) are observed at 30.96 ppm and 42.42 ppm, respectively. The peaks at 127.25 ppm, 133.01 ppm, 142.67 ppm, and 151.25 ppm are assigned to the phenyl carbons, and the chemical shift for the carbonyl carbon (-C=O) is observed at 165.26 ppm. The spinning sidebands for the phenyl carbons are marked with asterisks (*) in Figure 2. 

In the APS-OH CPI spectrum (Figure 2b), the ^13^C chemical shifts for the methyl, iso- propyl, phenyl, and carbonyl carbons are similar to those observed in the APS CPI spectrum. In addition, the chemical shift for the OH-bonded carbon is observed at 157.71 ppm. The chemical shifts of all carbons shown in Figure 2 are consistent with the synthesized CPI chemical structure [28,29].

### 3.3. Thermal Properties

The thermal properties of PVA, CPI, and the CPI blend films, synthesized using APS or APS-OH, containing varying amounts (wt%) of PVA, are listed in Table 2. The glass transition temperature (*T_g_*) of the APS CPI film shows an almost constant value at 204–207 °C, regardless of the PVA content (0–5.0 wt%). This is the similar to the APS-OH CPI, which exhibited a near-same *T_g_* value of 236–240 °C, regardless of the PVA wt% in the CPI blend. The *T_g_* value of the APS-OH CPI blend film was higher than that of the APS CPI, regardless of the PVA wt%. These values show that the -OH group of APS-OH increases the *T_g_* value of CPI by inducing intermolecular attractions between the CPI main chains through hydrogen bonding. These hydrogen bonds between the CPI main chains interfere with free chain mobility, resulting in an increase in *T_g_*. Similar papers have been previously published [30]. Figure 3 shows the DSC thermograms of the CPI blend films containing various PVA wt% contents.

The initial decomposition temperatures (*T_D_^i^*) of the CPI blend films containing APS and APS-OH for various amounts of PVA are summarized in Table 2. For the APS CPI, *T_D_^i^* showed a constant value in the range 463–468 °C, although the PVA content increased from 0 to 5.0 wt% (Table 2). This is similar to the APS-OH CPIs, i.e., as the amount of PVA increased up to 5.0 wt%, the *T_D_^i^* value remained in the range 328–335 °C. Figure 4 shows the TGA values of CPI films synthesized using monomers with two different structures. In the thermograms of the two CPI series according to the various amounts of PVA, the transition temperatures (*T_g_*, *T_m_*, and *T_D_^i^*) of PVA were not observed. These results imply that the water-soluble PVA in water was completely removed from the CPI blend film.

The residual amount (%) of the two CPI films at 600 °C (wt_R_^600^) showed an almost constant value as the PVA wt% increased to 5.0 wt%, i.e., the *wt_R_^600^* values of the APS CPI and APS-OH PI films were maintained at 52–56% and 65–67%, respectively. Because the thermal stability of the -OH group was low; therefore, the *T_D_^i^* value of APS-OH PI was lower than that of APS PI. On the contrary, due to the intermolecular hydrogen bonding of the PI main chain by the -OH groups, the *wt_R_^600^* of APS-OH CPI was higher than that of APS CPI. The *T_g_*, *T_D_^i^*, and *wt_R_^600^* values suggest that the PVA content had little effect on the thermal properties because the water-soluble PVA present in the PI film was removed by immersion in water.

In the TGA thermogram of Figure 4b, two-stage pyrolysis was observed at ~320 °C and ~420 °C, respectively. When -OH group is present in the polymer structure, PI is converted to polybenzoxazole (PBO), which has superior thermal stability through thermal rearrangement (TR) during heat treatment [31,32]. In many studies, PBO films processed by TR show abnormal microporous properties in the solid phase due to a large increase in free volume during TR. The PBO film obtained by heat treatment of APS-OH PI is an excellent material for gas separation applications, such as CO_2_/CH_4_ and CO_2_/N_2_ [33,34].

### 3.4. Membrane Morphology 

The XRD diffractograms of two types of CPI blend films containing varying amounts of PVA are shown in Figure 5. Peaks characteristic for PVA are generally observed at 2θ = 19.68° (d = 4.51 Å) and 23.64° (d = 3.75 Å). However, even if the amount of PVA increased from 0 to 5.0 wt% in all peaks of the APS and APS-OH CPI blends, no intrinsic peak of PVA was observed. In addition, the degree of crystallinity was almost constant, regardless of the amount of PVA contained in the CPI blend film (Table 2). These XRD patterns show that the PVA components present in the CPI blend film were completely removed at all PVA concentrations. The results of XRD are approximate results showing the dispersion of the filler and cross-checking with electron microscopy is required to fully confirm a particular dispersion state. 

SEM was used to confirm the various pore dispersions in the CPI microporous films after PVA removal. Figure 6 represents the SEM micrographs of the porous APS CPI films at various amounts of PVA. The pore size gradually increased according to the PVA concentration increased to 5.0 wt%, i.e., when the PVA concentration increased to 0.5 wt%, the average pore diameter was 0.21 μm, but, when the PVA wt% became 2.0, the pore diameter was ~0.66 μm. When the PVA concentration was 5.0 wt%, the average pore size increased to an average of 0.86 µm (see Table 2). Most of the pores shown in Figure 6 were evenly observed throughout the CPI film.

Figure 6 also shows the average size of the micropores according to the PVA concentration in the CPI blend film. When the PVA content increased up to 5.0 wt% in the CPI blend film, the PVA self-aggregates and the size of the particles increased constantly. When these PVA particles were dissolved and removed by water, the pore size of the CPI blend film increased accordingly. Table 2 shows the average size of the membrane pores according to the PVA concentration in the CPI blend film.

Similar results were observed for the APS-OH CPI, as listed in Table 2. The average pore diameter gradually increased as the amount of PVA increased up to 5.0 wt% in this CPI blend, e.g., when the PVA wt% increased from 0 to 1.0 in the CPI blend, the average pore diameter was observed to be 0.20 μm. However, when the PVA increased to 5.0 wt%, the pore diameter also increased with an average of 0.36 μm. Figure 7 shows the change in pore size according to the PVA concentration in the APS-OH CPI microporous film. 

The constant pore size of the CPI microporous film could be easily obtained by controlling the amount of PVA contained in the blend film. This result was possible not only with hydrophilic APS-OH monomer but also with non-hydrophilic APS monomer. The separation technique used to treat suspensions usually uses a microfiltration method, mainly filters with a porous membrane size of 0.01 to 1.0 μms. Therefore, the porous CPI film obtained in this study can filter out large portions of suspended particles, clays, bacteria or large viruses present in solution [35,36].

To compare the overall pore dispersion, SEM images of micro-pores in two CPI blend films containing the same amount of PVA are shown in Figure 8. After blending 5 wt% PVA with CPI, the average pore diameters of APS and APS-OH CPI microporous films with the PVA components removed by water were found to be approximately 0.84–0.88 µm and 0.34–0.38 µm, respectively. The overall pore size was constant and the dispersibility excellent. The average pore diameter of the APS-OH CPI film was smaller than that of the APS CPI film at the same PVA content. This is because a dense structure is formed by the main chains of CPI due to the hydrogen bonding of the -OH groups in APS-OH CPI. Therefore, when PVA is dissolved and eliminated from the PI blend film, the pore size of the APS-OH CPI film is smaller than that of the APS CPI film.

### 3.5. Optical Transparency

The optical transparencies of the CPI blend film can be determined by the yellow index (YI), cutoff wavelength (λ_o_), and transmittance at 500 nm wavelength (500^trans^ nm) [37,38,39]. The results obtained using UV-vis spectroscopy are displayed in Table 3 and Figure 9, respectively. For all APS and APS-OH PI blend films, regardless of the amount of PVA wt%, the λ_o_ value representing the initial transmittance was less than 400 nm, before the visible light region (see Table 3). These λ_o_ values are considerably lower compared to commercially available Kapton^®^ PI films, indicating the almost-transparent properties of these films. As shown in Table 3, the APS CPI blend film showed a maximum UV transmittance of 81–88% at 500 nm depending on the change in the content of PVA. In addition, the resulting value of the APS-OH CPI blend film significantly decreased from 83 to 18% as the amount of PVA increased to 5 wt%. The UV transmittance decreased as the content of PVA increased because the -OH groups in the PI main chain increased the bonding strength between the CPI chains due to intermolecular hydrogen bonding, resulting in the formation of dense structures. 

Table 3 lists the YI values of the APS CPI and APS-OH CPI blend films. The YI values of the APS CPI blend film were constant, regardless of all PVA content (0.5–5.0 wt%), and were very low overall (YI = 2–3). This value can be obtained from an almost colorless, transparent film, e.g., poly(methyl methacrylate) (PMMA) shows YI = 1–2 [40]. Conversely, for the APS-OH CPI blend film, the YI value steadily increased from 5 to 10 as the amount of PVA increased from 0 to 5 wt%. The reason is also the hydrogen bonding between the polymer chains. In order to confirm the actual colorless transparency, a 20–24-μm-thick film was prepared and tested with the naked eye. APS CPI showed very good optical transparency at all concentrations (Figure 10), but, for the APS-OH CPI film, when the PVA content increased up to 5 wt%, the yellow intensity further increased (Figure 11). However, both series of CPI films resulted in no difficulty reading logos through the film.

### 3.6. Solubility

The synthesized CPI blend films were examined for solubility in several solvents and the results are listed in Table 4. None of the films dissolved in the general-purpose solvents acetone, chloroform, alcohol, or toluene, and did not dissolve in DMAc, *N*-methyl-2-pyrrolidone (NMP), or dimethyl sulfoxide (DMSO), often used as solvents for PI. 

Because CPI is a highly functional polymer material with very strong chemical and heat resistance, its use is sometimes restricted to that of a general engineering material. However, the CPI blend film obtained in this study can be applied to the field of membranes using harsh, strong solvents, and the colorless, transparent membrane is another advantage.

## 4. Conclusions

PAA was synthesized by reacting two types of diamines, APS or APS-OH, with anhydrous BPADA. PAA was blended with various PVA concentrations and subjected to various heat treatment processes to produce CPI/PVA blend films. These films were immersed in water to remove water-soluble PVA to prepare two series of microporous CPI films. Because the APS-OH monomer used as an amine contains an -OH group in the PI main chain, the structure enabled hydrogen bonding between PI and PVA present in the blend, as well as between PI chains. The properties were compared with those of APS CPI without hydrogen bonds. Particularly in the case of APS-OH CPI, the pore size can be easily adjusted owing to its hydrophilicity. Therefore, APS-OH CPI can be used in MF membranes that require optical transparency and high functionality. 

The CPI microporous film developed in this study should be useful, not only as a filtration field but also as a functional polymer material because of its superior thermal properties, optical transparency, and solvent resistance compared to general engineering polymers. 

## Data Availability

The data presented in this study are available on request from the corresponding author.

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
