# Peer review of "Colorless and Transparent Polyimide Microporous Film with Excellent Physicochemical Property"

_polymers, 2021, doi:10.3390/polym13081298_

Round 1

Reviewer 1 Report

The work reported in this manuscript entitled (Colorless and Transparent Polyimide Membrane with Excellent Physicochemical Property) is interesting and well presented. However, it needs improvements before acceptance. The work requires minor revision.

Comment 1: There are some typographical errors in the manuscript, so authors need to correct them in the revised manuscript.

Comment 2: In Figures 7 and 8, The SEM images scale bar is not properly visible, so redraw the scale bar manually.

Comment 3: In SEM results: The authors should explore and discuss better their results with some more references in order to prepare a better discussion.

Comment 4: Conclusions should be shortened and extracted with the most important findings.

Reviewer 2 Report

1) I do not recommend to use the term "membrane" in the title of the manuscript because materials have not been tested in the role of membranes.

2) The dependences in the Figures 3 and 5 are not clear enough, so I recommend to adjust them.
